# Graphene Oxide Topical Administration: Skin Permeability Studies

**DOI:** 10.3390/ma14112810

**Published:** 2021-05-25

**Authors:** Filipa A. L. S. Silva, Raquel Costa-Almeida, Licínia Timochenco, Sara I. Amaral, Soraia Pinto, Inês C. Gonçalves, José R. Fernandes, Fernão D. Magalhães, Bruno Sarmento, Artur M. Pinto

**Affiliations:** 1i3S—Instituto de Investigação e Inovação em Saúde, Universidade do Porto, Rua Alfredo Allen, 208, 4200-180 Porto, Portugal; flsilva@i3s.up.pt (F.A.L.S.S.); rcalmeida@i3s.up.pt (R.C.-A.); saraamaral1998@gmail.com (S.I.A.); soraia.pinto@i3s.up.pt (S.P.); icastro@ineb.up.pt (I.C.G.); bruno.sarmento@ineb.up.pt (B.S.); 2INEB—Instituto de Engénharia Biomedica, Universidade do Porto, Rua Alfredo Allen, 208, 4200-180 Porto, Portugal; 3LEPABE—Laboratory for Process Engineering, Environment, Biotechnology and Energy, Faculdade de Engenharia, Universidade do Porto, 4200-180 Porto, Portugal; up201809122@fe.up.pt (L.T.); fdmagalh@fe.up.pt (F.D.M.); 4ICBAS–Instituto de Ciencias Biomédicas Abel Salazar, Universidade do Porto, Rua Jorge Viterbo Ferreira, 228, 4050-313 Porto, Portugal; 5CQVR–Centro de Química Vila Real, Universidade de Trás-os-Montes e Alto Douro, Quinta de Prados, 5001-801 Vila Real, Portugal; jraf@utad.pt; 6Physical Department, University of Trás-os-Montes and Alto Douro, Quinta de Prados, 5001-801 Vila Real, Portugal; 7CESPU, IINFACTS–Institute for Research and Advanced Training in Health Sciences and Technologies, Rua Central de Gandra 1317, 4585-116 Gandra, Portugal

**Keywords:** biocompatibility, carbon nanomaterials, graphite, phototherapy, skin disease

## Abstract

Nanostructured carriers have been widely used in pharmaceutical formulations for dermatological treatment. They offer targeted drug delivery, sustained release, improved biostability, and low toxicity, usually presenting advantages over conventional formulations. Due to its large surface area, small size and photothermal properties, graphene oxide (GO) has the potential to be used for such applications. Nanographene oxide (GOn) presented average sizes of 197.6 ± 11.8 nm, and a surface charge of −39.4 ± 1.8 mV, being stable in water for over 6 months. 55.5% of the mass of GOn dispersion (at a concentration of 1000 µg mL^−1^) permeated the skin after 6 h of exposure. GOn dispersions have been shown to absorb near-infrared radiation, reaching temperatures up to 45.7 °C, within mild the photothermal therapy temperature range. Furthermore, GOn in amounts superior to those which could permeate the skin were shown not to affect human skin fibroblasts (HFF-1) morphology or viability, after 24 h of incubation. Due to its large size, no skin permeation was observed for graphite particles in aqueous dispersions stabilized with Pluronic P-123 (Gt–P-123). Altogether, for the first time, Gon’s potential as a topic administration agent and for delivery of photothermal therapy has been demonstrated.

## 1. Introduction

Skin diseases are one of the leading causes of global disease burden, affecting millions of people worldwide. In the United States of America (USA), nearly 85 million people are seen by a physician for at least one skin disease every year. This leads to an estimated direct health care cost of USD 75 billion and an indirect lost opportunity cost of USD 11 billion. Further, mortality was noted in half of the 24 skin disease categories. The costs and prevalence of skin disease are comparable with or exceed other diseases with significant public health concerns, such as cardiovascular disease and diabetes. Chronic and incurable skin diseases, such as psoriasis, atopic dermatitis (AD), and vitiligo are associated with physical discomfort and impairment of patients’ quality of life whereas malignant diseases, such as basal cell carcinoma, if not early and properly treated can lead to mortality. Therefore, new and more effective treatment strategies are needed to deal with skin disease [1,2].

Guidelines for treatment of Psoriasis, AD, and Vitiligo include as first or second line options phototherapy with ultraviolet radiation combined or not with drugs (e.g., psoralen), while for basal cell carcinoma (BCC), photodynamic therapy (PDT) with photosensitizers, such as 5-aminolevulinic acid (5-ALA), activated by near-infrared (NIR) radiation are listed in the first line of treatments [3,4,5,6]. This therapy has also been reported in the literature for the abovementioned diseases [7]. However, such treatments still present limitations due to the low stability, toxicity, and skin penetration of commonly used drugs [8].

Nanostructured carriers are an upcoming option for drug delivery because of their advantages over conventional formulations. Nanoparticles can penetrate skin depending on size, charge, and surface chemistry. These colloidal particulate systems with sizes often around or below 200 nm offer targeted drug delivery, sustained release, improved biostability, and low toxicity. Nanoparticles are observed to penetrate skin intracellularly, intercellularly, or via hair follicles. Many nanocarriers such as polymeric, inorganic and lipid nanoparticles have been developed, and some like carbon based nanomaterials still need further exploration for future use in dermatological applications [9,10].

Carbon materials have been generally reported to have strong light absorption while maintaining their stability, and therefore being a promising new class of agents for phototherapy. Furthermore, such materials’ skin biocompatibility has been observed since ancient times, where manmade permanent tattoos using pulverized charcoal could be placed under the skin without apparent adverse effects [11].

A more recent type of carbon materials, graphene-based materials (GBM), have been widely explored as promising drug delivery vehicles. The large specific surface area of GBM facilitates efficient loading of drugs via surface adsorption or chemical functionalization. Graphene-based nanosystems have been shown to improve the stability, bioavailability, and photodynamic efficiency of organic photosensitizer molecules. They have also been shown to behave as electron sinks for enhanced visible light photodynamic activities. Owing to its intrinsic near infrared absorption properties, GBM can be designed to combine both photodynamic and photothermal hyperthermia for optimum therapeutic efficiency. Compared with other nanocarriers, GBM possess much higher drug loading capacity and radiation absorbance. It has been shown that GBM can be targeted to specific cells, for delivery of photosensitizers in PDT [12,13,14,15,16,17]. Furthermore, GBM are similar to active substances used in the treatment of dermatological conditions, such as psoriasis (e.g., anthracene, anthralin, psoralen, coal tar), which reaffirms the potential for phototherapeutic effect using GBM themselves and their good affinity with the drugs to be delivered. GBM have been shown to be biocompatible up to high concentrations that will hardly be achieved in dermatological phototherapy [18,19,20]. Moreover, some GBM have been reported to be biodegradable by human enzymes [21,22]. For those reasons, the use of GBM can be regarded as a promising option for target applications.

Graphene is the elementary structure of graphite and is composed of a single layer of sp^2^ hybridized carbon atoms organized in a hexagonal crystalline structure, forming a two-dimensional sheet. This material possesses high surface area, mechanical strength and thermal and electrical conductivity that supports its application in fields as diverse as energy technology, nanoelectronics, composite materials, and sensors [23,24,25,26,27,28,29,30,31]. In addition, graphene also possesses good optical transparency (97.7%) and high extinction coefficient in the NIR range, responsible for its high photothermal conversion ability [32].

The application of graphene in the biomedical area is limited by its hydrophobicity, which can be surpassed by its oxidation and consequent introduction of oxygen-containing functional groups, such as carboxyl, hydroxyl and epoxide groups [33]. Graphene oxide (GO) is similar to graphene, but the presence of these polar and reactive groups allows surface functionalization and coupling with other molecules such as chemotherapeutic drugs or photosensitizers that make possible its utilization as drug carriers. Thus, several biomedical applications of GO have also been studied, including biosensing/bioimaging, drug delivery, antibacterial or cancer photothermal therapy [34,35,36,37,38].

Beyond their polarity, materials’ size is of key importance in biomedicine. Considering that biological systems as membranes and protein complexes are natural nanostructures, the utilization of nanomaterials has a clear advantage in the interaction with these structures, making cellular uptake, penetration into blood vessels and renal clearance possible [39]. Thus, the successful application of GO in the biomedical field requires size reduction to the nanoscale.

The administration of nano graphene oxide (GOn) in in vivo models to test the efficacy of these materials as platforms for cancer or infections treatment, is generally done by intravenous or intratumoral injection [40,41,42,43,44]. However, these approaches present some disadvantages, once they are invasive procedures, more susceptible to trigger adverse reactions [45,46]. Thus, the topical application of GOn to treat skin diseases, including skin cancer, local infections or other diseases for which the treatment can be delivered through this route, is positioned as an interesting approach, since it is a non-invasive procedure that allows a localized material distribution, preventing any systemic side effects [45,46,47,48,49].

In view of these aspects, for the first time, to our knowledge, the permeability of single layer GO with nanometric lateral dimensions (GOn) and micrometric graphite stabilized with Pluronic P-123 (Gt–P-123) water dispersions through human skin have been studied. The influence of materials’ lateral dimensions and exfoliation procedure in skin permeation were also discussed. Finally, GOn photothermal therapy potential and biocompatibility were evaluated.

## 2. Materials and Methods

### 2.1. Graphite Dispersions Preparation

Graphite powder (size ≤ 20 µm, Sigma Aldrich, St. Louis, MO, USA) dispersions were stabilized with Pluronic P-123 (Sigma Aldrich, St. Louis, MO, USA). Graphite powder (Gt) and Pluronic P-123 (P-123) at final concentrations of 1000 µg mL^−1^ and 0.5% (*w*/*w*), respectively, were dispersed in deionized water and then sonicated for 10 min using an ultrasonic bath (ATM40-3LCD, Ovan, Barcelona, Spain) to obtain stable dispersions.

### 2.2. GOn Dispersions Production

Graphite oxide (GtO) was produced by Gt oxidation (size ≤ 20 µm, Sigma Aldrich, St. Louis, MO, USA) using the modified Hummers method, as described elsewhere [18,50]. Briefly, 4 g of graphite was added to a mixture of 40 mL of phosphoric acid (H_3_PO_4_, Chem-Lab, Zedelgem, Belgium) and 160 mL of sulfuric acid (H_2_SO_4_, VWR, Frankfurt, Germany) under stirring, and cooled using an ice bath. Then, 24 g of potassium permanganate (KMnO_4_, JMGS, Odivelas, Portugal) were added gently under stirring. Subsequently, 600 mL of H_2_O was slowly added, controlling temperature using an ice bath. Finally, hydrogen peroxide (H_2_O_2_, 26.5 mL, VWR, Frankfurt, Germany) was added and the mixture was left to rest overnight. Afterwards, the solution was decanted to separate the solid phase from the acidic solution, centrifuged at 4000 rpm for 20 min and redispersed in distilled water. The process was repeated until water pH was achieved in the supernatant. The pellet was recovered, redispersed in distilled water and sonicated for 8 h using a high-power ultrasonic probe (UIP1000hd, Hielscher Ultrasonics GmbH, Teltow, German) to simultaneously exfoliate GtO and breakup the sheets to lateral sizes close to a hundred nanometers, yielding the final product, nanographene oxide (GOn) at a concentration of 7 mg mL^−1^, which was further diluted for testing.

### 2.3. Characterization

#### 2.3.1. Optical Microscopy

Gt–P-123 dispersions at a Gt concentration of 1000 µg mL^−1^ were placed in a 48-well cell culture plate (500 µL) and observed under an inverted optical microscope (CKX41, Olympus, Tokyo, Japan) coupled with a digital camera (SC30, Olympus, Tokyo, Japan).

#### 2.3.2. Transmission Electron Microscopy

GOn sheets’ morphology and dimensions were evaluated using transmission electron microscopy (TEM, JEOL JEM 1400 TEM, Tokyo, Japan). An amount of 10 µL of GOn dispersed in water (50 µg mL^−1^) as placed on a carbon-coated TEM grid and left to stand for one minute. The surplus of the dispersion was removed using filter paper by capillarity. GOn lateral dimensions were measured from several different TEM images using ImageJ 1.53a software [51].

#### 2.3.3. Dynamic Light Scattering and Zeta Potential Measurements

The size of GOn particles, polydispersibility index (PDI) and values of zeta potential were assessed using a Zetasizer (Nano-ZS, Malvern Instruments, Malvern, UK) by dynamic light scattering (DLS) and electrophoretic light scattering (ELS). GOn (25 µg mL^−1^) was tested using a disposable Zetasizer cuvette (Malvern Instruments, Malvern, UK), at room temperature, and pH 6. Measurements were done in triplicate and results are presented as the average and standard deviation.

#### 2.3.4. Ultraviolet-Visible Spectroscopy

Absorption spectra in the range of 200–850 nm for GOn, G-P-123, and P-123 (only) were obtained using a spectrophotometer (Lambda 35 UV/Vis, Perkin-Elmer, MA, USA). Samples at 25 µg mL^−1^ concentration were analyzed in a 50 µL quartz cuvette (Hellma Analytics, Müllheim, Germany) with 10 mm light path length. All measurements were subjected to baseline correction using water as a blank control at room temperature.

### 2.4. Skin Permeation Experiments

#### 2.4.1. Human Samples

Human skin samples with a thickness of 0.8 mm were obtained from one healthy woman subjected to abdominal surgery in the Department of Plastic Surgery of the São João Hospital (Porto, Portugal). A written informed consent form was provided to the donor and the the Bioethics Committee of the São João Hospital approved the experimental protocol (protocol code: 90_17). The skin sample was washed with ultrapure water, and afterwards the hair and subcutaneous adipose tissue were removed using scissors. The skin was kept at −20 °C wrapped in aluminum foil until being used [52] as recommended by the European Center for the Validation of Alternative Methods, the International Programme on Chemical Safety and the EU Scientific Committee on Consumer Products.

#### 2.4.2. Skin Permeation Assays

Human skin permeability to Gt–P-123 and GOn was assessed using Franz diffusion cells with 9 mm clear jacketed with flat ground joint, 0.785 cm^2^ of permeation area, and with a receptor compartment with 5 mL of volume (PermeGear, Inc., Hellertown, PA, USA).

The skin previously prepared was mounted in the Franz cells with *stratum corneum* (SC) facing the donor compartment. The receptor chamber was filled with 0.1 M phosphate buffer (PBS) at pH 7.4, and maintained at 37 °C under stirring at 300 rpm, ensuring sink conditions. Afterwards, 500 µL of Gt (1000 µg mL^−1^) or GOn dispersions (300, 400, 500, and 1000 µg mL^−1^) were added to the donor compartment and sealed with paraffin film to ensure occlusive conditions in order to prevent loss of sample from the surface of the skin and also to maintain human skin hydrated during the assay [53]. Then, at 1, 2, 3, 4, 5 and 6 h, a receptor medium aliquot (100 µL) was recovered to determine by absorbance the amount of material that permeated through the skin. The same volume of PBS was then readded to the same compartment. A calibration curve for both materials was prepared to extrapolate Gt–P-123 or GOn concentrations at the receptor compartment. Materials’ permeated mass was obtained by multiplying the sample concentration for the volume of receptor compartment. Results are presented as cumulative mass and percentage of material that permeated through the skin. All assays were performed in triplicate.

### 2.5. GOn Photothermal Therapy Potential

In order to evaluate the ability of GOn to convert light into heat, 500 µL of the dispersion at different concentrations (300–1000 µg mL^−1^) was placed in a cell culture plate (48-well). A control was performed filling wells with water only. Samples irradiation was performed using a LED-based source with 150 mW cm^−2^ of irradiance and with a peak emission in NIR region (810 nm) [54]. Samples’ temperature increment induced by irradiation was monitored during 30 min using a type K thermocouple (Hanna instruments, Póvoa de Varzim, Portugal) placed at half-height and centered in the liquid. Assays were performed in 3 different experiments, with 3 replicates for each condition, and results are reported as the average and standard deviation of absolute temperature.

### 2.6. In Vitro Studies

#### 2.6.1. Cell Culture

HFF-1 human skin fibroblast cells (SCRC-1041, ATCC, Manassas, VA, USA) were utilized in the biological studies. Cells were cultured in Dulbecco’s Modified Eagle’s Medium (DMEM, ATCC, Manassas VA, USA) supplemented with 1% (*v*/*v*) penicillin/streptomycin (Biowest, Nuaillé, France) and 10% (*v*/*v*) fetal bovine serum (Alfagene, Lisbon, Portugal). Cells were kept at 37 °C in a humidified atmosphere with 5% CO_2_.

#### 2.6.2. Resazurin Assay

The effect of GOn on HFF-1 cells’ viability was assessed using different material amounts (180–600 μg/well, corresponding to 300–1000 μg mL^−1^). Each well has an area of 0.91 cm^2^. Cells at a density of 1 × 10^4^ cells/well were seeded in 48-well cell culture plates and incubated for 24 h at 37 °C and 5% CO_2_. Afterwards, cell medium was removed and GOn dispersions were added in a final volume of 600 µL per well (in complete DMEM). After, 24 h incubation, cell viability was quantified by using the resazurin assay. GOn dispersions were removed, cells were washed 3 times with PBS and then incubated at 37 °C and 5% CO_2_ for 2 h with 10% (*v*/*v*) resazurin (Sigma-Aldrich, St. Louis, MO, USA) previously prepareded in cell culture medium. The supernatant fluorescence (λ_ex/em_ = 530/590 nm) of each well was determined using a Synergy Mx micro-plate reader (Bio-Tek Instruments, VT, USA). Cell viability decrease positive and negative controls were performed incubating HFF-1 cells with 10% (*v*/*v*) dimethyl sulfoxide (DMSO) in complete DMEM and complete DMEM only, respectively. Results for each condition were normalized to the negative control (cells in complete DMEM only) and reported as % of the control. All experiments were performed in triplicate and six replicates for each condition were performed.

#### 2.6.3. Live/Dead Assay

The effecf of GOn on cells morphology and viability was evaluated by performing a live/dead assay. Cells were seeded and exposed to GOn as described for the resazurin assay. After 24 h, cells were washed 3 times with PBS and incubated with calcein (1 µM) and propidium iodide (2 µg mL^−1^) in PBS during 15 min at 37 °C in the dark. Then, cells were washed twice wih PBS and analyzed using an inverted fluorescence microscope (Axiovert 200, Zeiss, Jena, Germany).

### 2.7. Statistical Analysis

Statistical analyses were performed using GraphPad Prism software version 8.4.2 (GraphPad Software, San Diego, CA, USA). One-way analysis of variance (ANOVA) with Tukey’s test for multiple comparisons were performed. Differences between experimental groups are considered significant whenever *p* < 0.05.

## 3. Results and Discussion

### 3.1. Gt and GOn Dispersions Physico-Chemical Characterization

Graphite (Gt) was dispersed by sonication in water, however, it precipitated due to its large size (≤20 µm) and hydrophobicity. Therefore, it was stabilized with Pluronic P-123 (P-123), a non-ionic surfactant composed of poly(ethylene oxide) and poly(propylene oxide) blocks [55]. Nanosized GO (GOn) was produced by Gt oxidation and exfoliation using a modified Hummers method, followed by high-power ultrasonication. Figure 1 shows Gt, Gt–P-123, and GOn aqueous dispersions at a concentration of 1000 µg mL^−1^.

The presence of P-123 at a concentration of 0.5% (*w*/*v*) stabilized Gt in water, allowing the attainment of a homogenous blackish dispersions without formation of any precipitate. Such dispersions are stable for a 12 h period, after which the sedimentation becomes visible. However, it is possible to easily redisperse them by manual shaking. GOn water dispersions presented a typical brownish appearance and good stability. Such dispersions present a shelf-life of at least 6 months (longest observation period tested).

Gt–P-123 water dispersions were observed by optical microscopy (Figure 2), being revealed to have small particles with sizes from a few µm to large agglomerates up to 200 µm. 

The morphology and size of GOn nanosheets were evaluated by TEM. Figure 3 shows that our high-power sonication size reduction metod allowed achievement of well exfoliated GOn single layer particles with an average size of 190 ± 144 nm. Furthermore, 70% of the particles measured presented sizes below 200 nm (Figure 3B), 90% were under 290 nm, and all particles measured presented lateral sizes below 450 nm. In addition, no agglomeration was observed (Figure 3A), confirming the good dispersibility and degree of exfoliation of the GOn particles.

Table 1 shows particle size, polydispersity index (PDI) and surface charge of GOn measured using a Zetasizer by DLS and ELS, respectively. Gt–P-123 dispersions presented a particle size too large to be analyzed using a Zetasizer; however, it has already been clearly observed in optical microscopy images (Figure 2), furthermore, this is a commercial material whose particle size has already been described. GOn presented hydrodynamic diameters of 197.6 ± 11.8 nm, with a PDI of 0.396 ± 0.013. Particle size average value determined by DLS for GOn is consistent with TEM measurements. The smaller size distribution range observed might have to do with particles being stabilized and folded due to intraplanar hydrophilic interactions when well dispersed in water, opposing to when adsorbed on the TEM grid’s surface [56]. The surface charge was −39.4 ± 1.8 mV, which is a high value that explains the excellent aqueous dispersion stability visually observed for more than 6 months for this material [57].

The absorbance spectra of GOn, Gt–P-123, and P-123 were determined by UV/Visible spectroscopy (Figure 4). GOn spectra presented an absorbance peak at λ_max_ = 230 nm, attributed to π–π* electronic transitions in *sp*^2^ clusters, and a shoulder peak at 300 nm, corresponding to n–π* transitions of free electron pairs in oxygen atoms in C=O bonds from carboxyl and carbonyl groups [51]. Gt–P-123, presented a typical spectrum for graphitic materials, with peaks at 223 and 273 nm [58]. Residual absorbance was detected for Pluronic-only when at the same concentration used to stabilize Gt in water.

### 3.2. Skin Permeability of GOn and Gt–P-123

The permeation through human skin of GOn and Gt–P-123 was evaluated immobilizing the skin samples between the donor and receiver compartments of Franz cells (Figure 5A). The donor compartment was filled with 500 µL of GOn (300–1000 µg mL^−1^) or Gt–P-123 (1000 µg mL^−1^). Samples were collected from the receptor compartment every hour, for 6 h. The amount of material that permeated the skin was quantified by UV/Visible spectroscopy. GOn and Gt–P-123 concentrations were determined from absorbance values at wavelengths correspondent to the maximum absorption peaks in their spectra (230 nm for GOn and 223 nm for Gt–P-123). This was performed by matching the absorption values obtained with calibration curves performed with a range of known concentrations of both materials.

Gt–P-123 was not detected in the receptor compartment even after 6 h, indicating that it cannot permeate through the skin sample. This reaffirms the relevance of reaching nanometric size to achieve and maximize skin permeation of nanoparticles [59]. This material has therefore no use as a possible vehicle for drug delivery or phototherapy in skin diseases, and was not further characterized.

Results for GOn skin permeation are presented in Figure 5B. GOn was capable of permeating across the skin in a time-dependent manner for all concentrations tested. It is relevant to notice that, besides presenting a lateral size below 200 nm, GOn is formed by a single layer of carbon atoms, therefore presenting a very low thickness and high flexibility, which facilitates transport through skin. On the other hand, Gt is composed of numerous stacked graphene layers.

Skin permeability of GOn at concentrations of 300, 400, 500, and 1000 µg mL^−1^ was evaluated every hour for a period of 6 h (Figure 5B). The cumulative percentage of GOn that permeated from the donor to the receptor compartment was found to decrease as concentration increased. After 6 h in contact with skin 55.3, 91.4, 99.3 and 99.8% of the GOn placed in the donor compartment at 1000, 500, 400 and 300 µg mL^−1^, respectively, reached the receptor compartment. For high concentrations, permeation is hindered by deposition of larger particles and agglomerates along time.

After 1 h of experiment, the percentages of GOn at 1000 and 500 µg mL^−1^ at the receptor compartment were of 18.4 and 20.2%, respectively. After 3 h, permeation values were of 42.6 and 54.0%, in the same order. After 4 h, the permeation of GOn at 1000 µg mL^−1^ started to stabilize due to surface deposition and agglomeration. Values observed were of 49.0% and 61.6% for GOn at 1000 and 500 µg mL^−1^, respectively. After 6 h, the percentage of permeation obtained at 1000 µg mL^−1^ was 1.65-, 1.79- and 1.80-fold lower than when using GOn at 500, 400 and 300 µg mL^−1^. However, using GOn at 1000 µg mL^−1^ allowed the achievement of a higher absolute mass of material in the receptor compartment (276.7 µg), when compared to GOn at 500 (228.5 µg), 400 (198.6 µg) and 300 µg mL^−1^ (150 µg).

### 3.3. GOn Photothermal Therapy Potential

Since GOn particles’ ability to permeate through human skin has been demonstrated, they might have the potential to be used in dermatological applications, such as photothermal therapy of skin cancer [13,14,15,16,60]. For that reason, the ability of GOn (300–1000 µg mL^−1^) to convert NIR light into thermal energy was evaluated (Figure 6). GOn heating by NIR light irradiation was demonstrated to be concentration- and time-dependent. At a concentration of 300 µg mL^−1^, GOn reached temperatures of 36.0 and 40.2 °C, after 15 and 30 min of irradiation, respectively. For 400 and 500 µg mL^−1^, similar values were obtained of around 38 and 42 °C, after 15 and 30 min. Finally, at a concentration of 1000 µg mL^−1^, GOn dispersions reached temperatures of 40.3 °C and 45.7 °C, after 15 and 30 min of NIR irradiation, respectively. These values corresponded to an increment of up to 10 °C in relation to water only (control). Therefore, GOn dispersions confirmed to be effective agents to induce a temperature increase within the mild photothermal therapy temperature range, which has been reported to induce death of skin cancer cells [54,60]. Concentrations and times applied can be adjusted according to the specific patient and desired treatment.

### 3.4. In Vitro Biocompatibility of GOn

Since GOn has the potential to be used for applications such as skin cancer phototherapy and topic drug delivery [13,14,15,16,54,61], it is important to assure that the used particles are non-toxic towards healthy skin cells. For that reason, human foreskin fibroblasts (HFF-1) were incubated with increasing concentrations (300–1000 μg mL^−1^) of GOn for 24 h, and cell viability assessed through the resazurin assay (Figure 7).

It is relevant to mention that, unlike what happens with HFF-1 exposed to the full amount of material placed in the wells during 24 h, in skin permeation tests, the particles go through the skin in a period up to only 6 h. Therefore, the time of exposure and GOn amounts inside the skin are lower during permeation than in the in vitro biological tests presented in this section. GOn did not induce any statistically significant decrease in HFF-1 cell viability, for all conditions tested, as compared to the control condition in which the cells were incubated in cell culture media without materials. Figure 8 shows that HFF-1 cells presented a normal spindle like shape, characteristic of human skin fibroblasts, when exposed or not to GOn. Moreover, no death cells, stained with PI, could be found. This reaffirms the potential of the nanosized single layer GOn herein reported to be used in the biomedical field, in applications such as, for example, skin cancer phototherapy or topic drug delivery [13,14,15,16,54,61].

## 4. Conclusions

In order to evaluate their potential for dermatological applications, two different carbon materials were studied in terms of physicochemical characteristics and human skin permeation. Graphite particles in aqueous dispersions stabilized with Pluronic P-123 (Gt–P-123) presented sizes between a few to hundreds (agglomerates) of microns. The presence of P-123 at a concentration of 0.5% (*w*/*v*) stabilized Gt in water, allowing the attainment of homogenous blackish dispersions without sedimentation. Such dispersions are stable for 12 h, a period after which they precipitate. However, they can be easily redispersed. Gt–P-123 presented a typical spectrum for graphitic materials, with peaks at 223 and 273 nm. Due its large size, no skin permeation was observed for Gt–P-123.

Nanographene oxide (GOn) particles presented average lateral sizes of 197.6 ± 11.8 nm, and a surface charge of −39.4 ± 1.8 mV, being stable in water dispersion for up to 6 months. GOn spectra presented an absorbance peak at λ_max_ = 230 nm, attributed to π–π* electronic transitions in *sp*^2^ clusters, and a shoulder peak at 300 nm, corresponding to n–π* transitions of free electron pairs in oxygen atoms in C=O bonds from carboxyl and carbonyl groups.

GOn was capable of permeating across skin in a time-dependent manner. An amount of 20.3% of the mass of GOn (1000 µg mL^−1^) put in contact with the skin sample permeated after 1 h, while 55.5% permeated after 6 h. Lower concentrations of GOn (300–500 µg mL^−1^) presented faster permeation to the receptor compartment, however, the total mass of material that permeated was lower. Furthermore, GOn dispersions were shown to absorb near-infrared radiation, causing local temperature to reach up to 45.7 °C, within mild photothermal therapy temperature range. Concentrations and times to apply can be adjusted according to a specific patient and desired treatment.

Finally, GOn in amounts superior to those which could permeate the skin were shown not to affect human skin fibroblast (HFF-1) morphology or viability, after 24 h of incubation.

GOn potential as a topic administration agent and for delivery of photothermal therapy has been demonstrated. This material can also be considered as a drug delivery vehicle for drugs used in skin disease, potentially improving drugs’ stability and penetration, allowing for reduced therapeutic doses and avoiding side effects of systemic therapy and high topical doses.

## Figures and Tables

**Figure 1 materials-14-02810-f001:**
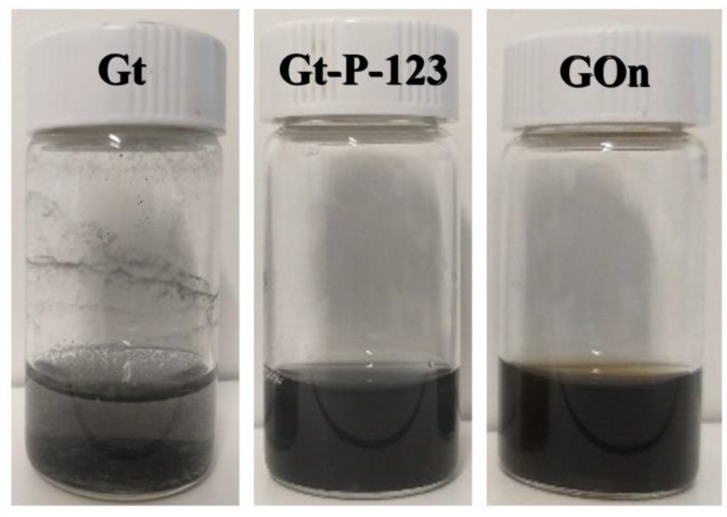
Images of graphite without Pluronic P-123 (Gt), Graphite + Pluronic P-123 (Gt–P-123) and graphene oxide (GOn) dispersions, at a concentration of 1000 µg mL^−1^, in glass vials for stability evaluation. Sedimentation is visible on the Gt sample after only a few seconds.

**Figure 2 materials-14-02810-f002:**
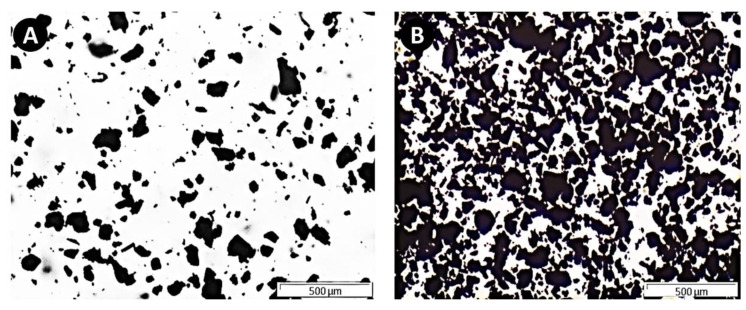
Representative optical microscopy images of Graphite–P-123 dispersions (1000 µg mL^−1^) at the edge (**A**) and center (**B**) of the well. The scale bar represents 500 µm.

**Figure 3 materials-14-02810-f003:**
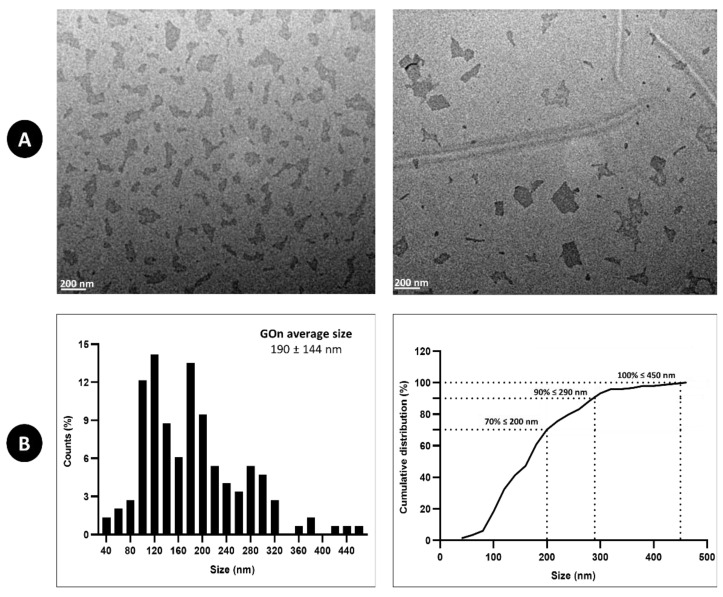
GOn particle morphology and size. (**A**) Representative TEM images of GOn aqueous dispersions (50 µg mL^−1^). Scale bar represents 200 nm. (**B**) GOn particle size distribution determined from TEM images.

**Figure 4 materials-14-02810-f004:**
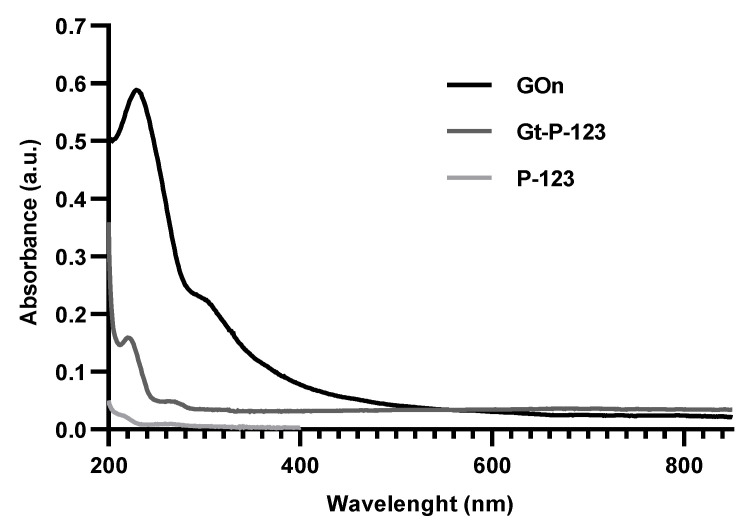
UV/Visible absorption spectra for water dispersions of nanographene oxide (GOn), graphite stabilized with Pluronic P-123 (Gt–P-123), and Pluronic P-123 only (P-123) (at the same concentration used to stabilize Gt).

**Figure 5 materials-14-02810-f005:**
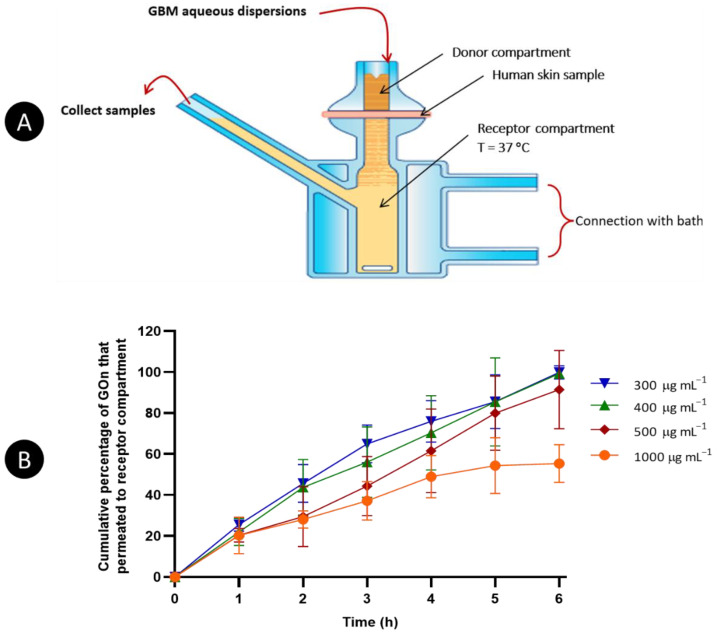
Skin permeation studies for GOn. (**A**) Schematic representation of the skin permeability experimental setup using a Franz cells system. (**B**) Cumulative percentage of GOn that permeated from donor to receptor compartment at different concentrations (300–1000 µg mL^−1^). Results are presented as the average and standard deviation.

**Figure 6 materials-14-02810-f006:**
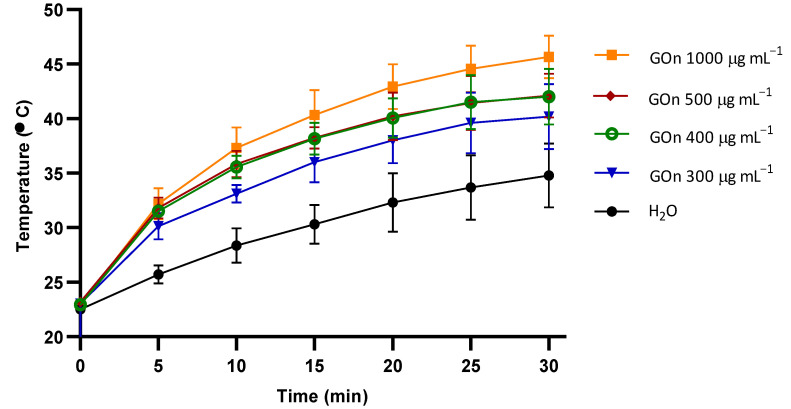
Photothermal heating curves of water-only or GOn aqueous dispersions at concentrations of 300, 400, and, 1000 µg mL^−1^.

**Figure 7 materials-14-02810-f007:**
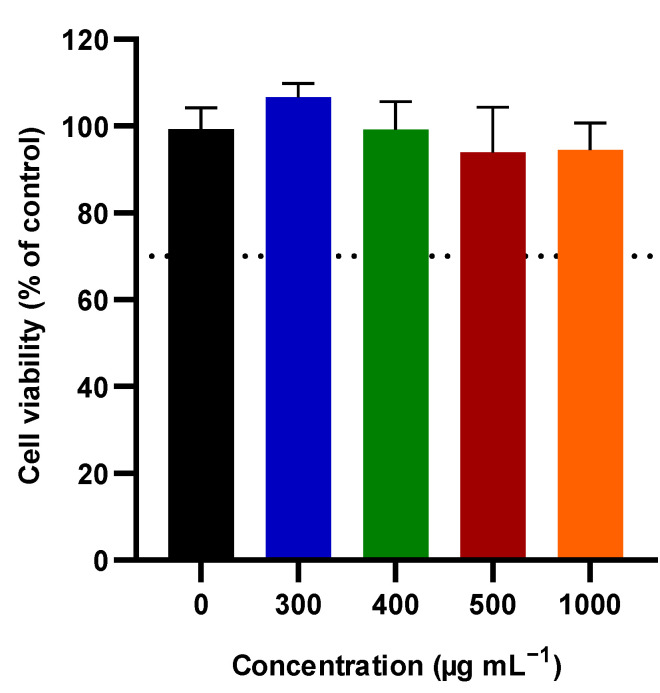
Cellular viability of HFF-1 cells determined using the resazurin assay. Results are normalized with respect to values of the control without material (cell culture media only), and presented as average and standard deviation. The dotted line at 70% marks the toxicity limit, according to ISO 10993-5:2009(E). No statistically significant differences were found between conditions tested.

**Figure 8 materials-14-02810-f008:**
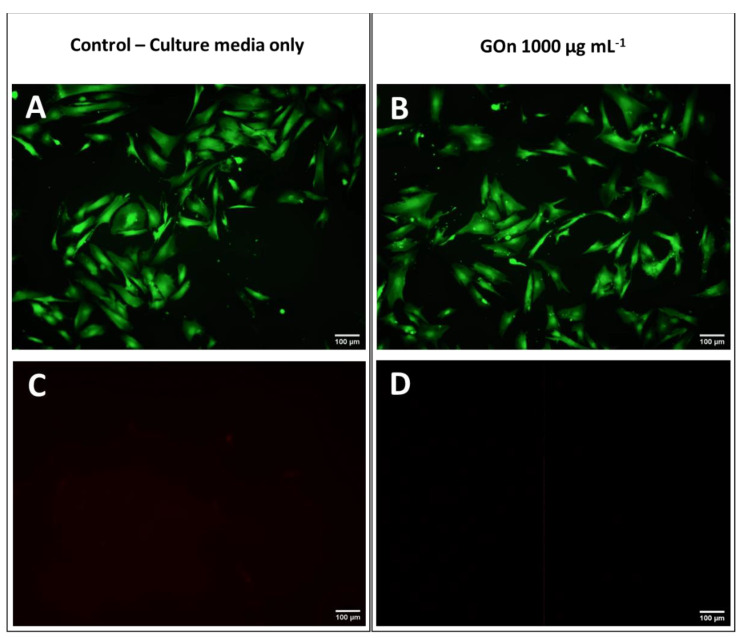
Representative fluorescence microscopy images of HFF-1 cells in cell culture media only (**A**,**C**) and after 24 h of contact with 1000 µg mL^−1^ of GOn (**B**,**D**). Live cells (**A**,**B**) are green (calcein) and dead cells (**C**,**D**) appear in red (propidium iodide). Scale bar represents 100 µm.

**Table 1 materials-14-02810-t001:** Size, surface charge and polydispersity index of GOn aqueous dispersions diluted at a concentration of 25 μg mL^−1^ and pH 6 (*n* = 3).

Material	Size (nm)	Polydispersity Index	Surface Charge (mV)
GOn	197.6 ± 11.8	0.396 ± 0.013	−39.4 ± 1.8

## Data Availability

The data presented in this study are available on request from the corresponding author.

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
