# Peer review of "Graphene Oxide Topical Administration: Skin Permeability Studies"

_materials, 2021, doi:10.3390/ma14112810_

Round 1

Reviewer 1 Report

The manuscript by Silva et al. focuses on GOn potential as a topic administration agent and carrier for delivery of photothermal therapy. It represents an interesting and decent study. However, I have some doubts and remarks to be considered:

1/ Materials and methods - please explain the rationale for skin samples storage at -20°C. Does it maintain the skin structure and unchanged permeability?

2/ Figures captions should be more informative, especially for Fig. 2 and 3. What are the differences between presented samples in Fig. 2 and 3? Please, complete them.

3/ Cell viability results are very scarce (Fig. 7), why did authors check only 3 concentrations of nGO, actually quite similar? Other experiments with nGO were performed in concentrations up to 1 mg/ml.

Author Response

The authors appreciate the reviewer comments which are very pertinent and helped to improve the manuscript.

Question 1: Materials and methods - please explain the rationale for skin samples storage at -20°C. Does it maintain the skin structure and unchanged permeability?

Reply: After obtaining human skin from abdominal surgery of a healthy patient, the subcutaneous fat and hair should be removed and the skin needs to be stored at -20 °C until further use. Previous studies already demonstrated that storage in such conditions for several months did not induce significant changes in the barrier function. Moreover, the European Center for the Validation of Alternative Methods, the International Programme on Chemical Safety and the EU Scientific Committee on Consumer Products strongly recommend the storage of human skin at -20 °C. Additional information regarding the previous was added in “Materials and Methods” section of the manuscript.

Question 2: Figures captions should be more informative, especially for Fig. 2 and 3. What are the differences between presented samples in Fig. 2 and 3? Please, complete them.

Reply: Figure 2 and 3 present representative images of graphite stabilized by Pluronic P-123 (Gt-P-123) dispersions obtained by optical microscopy or nano graphene oxide (GOn) obtained by transmission electron microscopy, respectively. Gt-P-123 particles exhibit sizes from a few µm, to large agglomerates up to 200 µm, thousand-fold higher than GO nanosheets. In addition, GOn distinguishes itself from Gt-P-123 dispersions by the absence of agglomerates. Extra data were added to Figure 3 and to the respective caption, as well as to the correspondent text in “Results” section.

Question 3: Cell viability results are very scarce (Fig. 7), why did authors check only 3 concentrations of nGO, actually quite similar? Other experiments with nGO were performed in concentrations up to 1 mg/ml.

Reply: Indeed, it would be relevant to evaluate the biocompatibility of GOn at a concentration of 1000 µg mL-1. All concentrations evaluated in skin permeability tests and for photothermal effect were now studied using the resazurin assay and results reveal that they are non-toxic towards human skin fibroblasts. Furthermore, a live/dead staining was performed, revealing that all cells exposed to GOn were alive after a period of 24h. All details about mentioned experiments were added to the manuscript.

Reviewer 2 Report

This is a really interesting work by the authors. My only two comments are that:

  1. The polydispesity index of the dispersions should be included. Homogeneity of the sample is really important.
  2. It would also be interesting to check if after permeating the epidermis the physical characteristics of the dispersion change (e.g. size etc) that would also affect the phototherapy efficiency.

Author Response

We thank the reviewer for his kind considerations about the interest of the work we have submitted to be evaluated for publication at Materials journal.

Question 1: The polydispesity index of the dispersions should be included. Homogeneity of the sample is really important.

Reply: The polydispersity index (PDI) value was added to the manuscript. GOn dispersions presented a PDI of 0.396 ± 0.013, indicating the narrow and homogeneous particle size of the materials when dispersed in water.

Question 2: It would also be interesting to check if after permeating the epidermis the physical characteristics of the dispersion change (e.g. size etc) that would also affect the phototherapy efficiency.

Reply: We have tried to evaluate the size of the particles after the permeation assays, however, because the materials are extensively diluted after crossing to the receptor compartment, that was not possible to be attained. It would be indeed an interesting characterization to perform and we thank the reviewer for the suggestion, but unfortunately it could not be attended due to the limited time available to provide the revised version of the manuscript. We will take it into consideration and seek ways of performing such characterization in future works.

Reviewer 3 Report

In the work titled “Graphene oxide topical administraton: skin permeability studies” authors presented solid results on the permeation ability of a GO dispersion through human skin.  They synthesized the GO sample using the modified Hummers method obtaining flakes below 200 nm size (GOn), and characterized the nanomaterial by TEM, DLS, Zeta potential and UV-vis. Moreover, they tested the PTT capacity of the dispersions and the cytotoxicity effect on fibroblasts. Finally, the authors compared the skin permeation capacity of GOn and the stabilized Gt-P-123 in water using Franz cells, concluding no skin permeation for Gt-P-123 due to its large size, contrary to GOn, which showed a permeability across skin in a time-dependent manner.

In conclusion, the work brings interesting results in the context of graphene derivatives as topical administration agents for skin related issues.

I recommend this manuscript for publication in Materials after addressing major issues.

  1. A GOn size distribution using TEM images, for instance, is necessary to perfectly characterize the size of the flakes. DLS is not appropriate for non-spherical particles. Indeed, most of the flakes observed on Figure 3 are much smaller than the DLS average value provided in Table 1.
  2. The authors should indicate why the used only one human skin sample. It is well known that skin characteristics vary depending on donor´s age, for instance.
  3. Some more specific information is missing in the experimental part for skin permeation assays. The authors should explain the importance of providing occlusive conditions. In addition, it is highly recommendable to perform the skin permeation studies using other GOn concentrations, for instance 300-500 ug/mL, which is the concentrations range chosen for the in vitro biocompatibility of GOn, in order to observe a dose-dependent permeation profile.
  4. Most probably, lower concentrations of GOn would also influence the photothermal heating curves. Did the authors tried also to irradiate the skin after GOn permeation in order to observe a PTT effect on the tissue?
  5. Authors present cell viabilities higher than 100% for fibroblasts when incubated with GOn in Figure 7. How can they justify these results? Other tools such as LDH assay could provide more accurate cytotoxicity results.
  6. It would be highly recommendable to perform a histology examination on the skin sample after GOn permeation, using also a “control” skin sample. The authors should clarify why, after 6h of exposure, they find a “plateau” in the cumulative mass of GOn and not all the nanomaterial but a 55.5% of the GOn, permeated the skin.

Author Response

The authors would like to thank the reviewer for his comprehensive analysis of the manuscript and for the suggestions that contributed to improve the work presented.

Question 1: A GOn size distribution using TEM images, for instance, is necessary to perfectly characterize the size of the flakes. DLS is not appropriate for non-spherical particles. Indeed, most of the flakes observed on Figure 3 are much smaller than the DLS average value provided in Table 1.

Reply: GOn size distribution was determined by TEM. Particle size average value determined by DLS for GOn (197.6 ± 11.8 nm) is consistent with TEM measurements (190 ± 144 nm). The smaller size distribution range observed might have to do with particles being stabilized and folded due to intraplanar hydrophilic interactions when well dispersed in water, opposing to when adsorbed on the TEM grid’s surface. All details about mentioned experiments were added to the manuscript.

Question 2: The authors should indicate why the used only one human skin sample. It is well known that skin characteristics vary depending on donor´s age, for instance.

Reply: Due to Covid-19 pandemic, all non-urgent surgeries were suspended in Portugal for undetermined time, which strongly limits the acquisition of skin samples. For this reason, we used skin samples available from one healthy donor this time. It is in fact worth to note the relevancy of testing skin from different donors, which we will perform as soon as the shortage of skin samples is normalized.

Question 3: Some more specific information is missing in the experimental part for skin permeation assays. The authors should explain the importance of providing occlusive conditions. In addition, it is highly recommendable to perform the skin permeation studies using other GOn concentrations, for instance 300-500 ug/mL, which is the concentrations range chosen for the in vitro biocompatibility of GOn, in order to observe a dose-dependent permeation profile.

Reply: The occlusive conditions are important to prevent loss of sample from the surface of the skin by evaporation, and also to maintain human skin hydrated during the assay, avoiding it from drying out, which explains the procedure employed in our study. Such information was added to the “Materials and Methods” section of the manuscript.

Skin permeability for GOn at concentrations of 300, 400 and 500 µg mL-1 were determined. Methods and Results were included in the manuscript and described as follows:

“Skin permeability of GOn at concentrations of 300, 400, 500, and 1000 µg mL-1 was evaluated every hour during a period of 6 h (Figure 5B). The cumulative percentage of GOn that permeated from the donor to the receptor compartment was found to decrease as concentration increases. After 6 h in contact with skin 55.3, 91.4, 99.3 and 99.8 % of the GOn placed in the donor compartment at 1000, 500, 400 e 300 µg mL-1, respectively, reached the receptor compartment. For high concentrations, there is higher material deposition and agglomeration, as time increases, at the surface of the skin.

After 1 h of experiment, the percentages of GOn at 1000 and 500 µg mL-1 at the receptor compartment were of 18.4 and 20.2 %, respectively. After 3 h, permeation values were of 42.6 and 54.0 %, in the same order. After 4 h, the permeation of GOn at 1000 µg mL-1 started to stabilize due to surface deposition and agglomeration. Values observed were of 49.0 % and 61.6 % for GOn at 1000 and 500 µg mL-1, respectively. After 6 h, the percentage of permeation obtained at 1000 µg mL-1 was 1.65, 1.79 and 1.80- fold lower than when using GOn at 500, 400 and 300 µg mL-1. However, using GOn at 1000 µg mL-1 allowed to achieve a higher absolute mass of material in the receptor compartment (276.7 µg), when compared to GOn at 500 (228.5 µg), 400 (198.6 µg) and 300 µg mL-1 (150 µg).”

Question 4: Most probably, lower concentrations of GOn would also influence the photothermal heating curves. Did the authors tried also to irradiate the skin after GOn permeation in order to observe a PTT effect on the tissue?

Reply: Photothermal heating curves for GOn at all concentrations tested were included in the manuscript and respectively described. GOn heating depends on irradiation time and material concentration. We thank the reviewer for the suggestion and we are working on implementing a system for irradiating the skin during the permeation tests, however, due to the complexity of the task it was not ready in time to meet the deadline for resubmitting the manuscript.

Question 5: Authors present cell viabilities higher than 100% for fibroblasts when incubated with GOn in Figure 7. How can they justify these results? Other tools such as LDH assay could provide more accurate cytotoxicity results.

Reply: Regarding results presented in Figure 7, indeed there is one value for GOn at a concentration of 300 µg mL-1 slightly higher than 100 %, however, this is within experimental error, having no biological meaning. For all concentrations tested cell viability is above the 70% toxicity limit defined by ISO 10993-5:2009(E). Also, no statistically significant differences were found between conditions tested. To further attest the validity of our conclusions regarding GOn non-toxicity in concentrations up to 1000 µg mL-1, a live/dead staining was performed, revealing that no dead cells stained with propidium iodide were present after a period of 24 h. All details about mentioned experiments were added to the manuscript.

Question 6: It would be highly recommendable to perform a histology examination on the skin sample after GOn permeation, using also a “control” skin sample. The authors should clarify why, after 6h of exposure, they find a “plateau” in the cumulative mass of GOn and not all the nanomaterial but a 55.5% of the GOn, permeated the skin.

Reply: It can be visually observed that for a concentration of 1000 µg mL-1, there is higher material deposition and agglomeration, as time increases, at the surface of the skin. This leads to a “plateu” in the total percentage of GOn that permeates through skin after 4 h of the assay. Such observations were included in the “Results” section, together with permeation values for lower concentrations, namely GO at 300, 400, and 500 µg mL-1.

Round 2

Reviewer 1 Report

The Authors have replied to all my concerns and corrected the suggested data. After the revision made the manuscript looks good enough for acceptance.

Reviewer 3 Report

The authors have properly answered to all the comments and they have improved the manuscript accordingly. The work is now suitable for publication in the present form in Materials journal.